# Opto-magnetic capture of individual cells based on visual phenotypes

**Loïc Binan[1,2], François Bélanger[1,3], Maxime Uriarte[1,3], Jean François Lemay[1], Jean Christophe Pelletier De Koninck[1], Joannie Roy[1], El Bachir Affar[1,3], Elliot Drobetsky[1,3], Hugo Wurtele[1], Santiago Costantino[1,2]\***

[1]Research Center, Maisonneuve-Rosemont Hospital, Montreal, Canada;
[2]Department of Ophthalmology, University of Montreal, Montreal, Canada;
[3]Department of Medicine and Molecular Biology Program, University of Montreal, Montreal, Canada

**Abstract** The ability to isolate rare live cells within a heterogeneous population based solely on visual criteria remains technically challenging, due largely to limitations imposed by existing sorting technologies. Here, we present a new method that permits labeling cells of interest by attaching streptavidin-coated magnetic beads to their membranes using the lasers of a confocal microscope. A simple magnet allows highly specific isolation of the labeled cells, which then remain viable and proliferate normally. As proof of principle, we tagged, isolated, and expanded individual cells based on three biologically relevant visual characteristics: i) presence of multiple nuclei, ii) accumulation of lipid vesicles, and iii) ability to resolve ionizing radiation-induced DNA damage foci. Our method constitutes a rapid, efficient, and cost-effective approach for isolation and subsequent characterization of rare cells based on observable traits such as movement, shape, or location, which in turn can generate novel mechanistic insights into important biological processes.
DOI: https://doi.org/10.7554/eLife.45239.001

**\*For correspondence:**
santiago.costantino@umontreal.ca

**Competing interests:** The authors declare that no competing interests exist.

## Introduction

Characterization of biological samples relies heavily on microscopy where, in response to various stimuli, molecular probes and a myriad of contrast reagents are routinely used to identify and label individual live cells of interest. These methods often require prior knowledge of cellular markers or use of elaborate reporter constructs. On the other hand, based solely on visual inspection or using image processing algorithms, it is possible to distinguish rare cells which exhibit distinct biological properties from among thousands of counterparts within a microscopy field. Such visually discernable traits include movement, shape, intracellular protein distribution, and location within the sample, and in turn can reflect important physiological features of individual cells. For example, cell migration (movement) is an essential determinant in normal embryonic development, wound healing, immune responses, tumor progression, and vascular disease (*Kurosaka and Kashina, 2008*). Moreover, changes in cellular morphology (shape) constitute biomarkers of cellular growth, division, death, and differentiation, as well as of tissue morphogenesis and disease (*Prasad and Alizadeh, 2019*). Cell-to-cell contacts (location) or distance to sources of chemical cues such as senescent cells, inflammation or necrotic tissue are critical factors in chemokinesis, differentiation, neural function, and immune responses (*Garcia et al., 2018*). Finally, expression and visualization of fluorescent fusion proteins permits the identification of cells presenting molecular behaviors of interest, such as differential relocalization of proteins to subcellular compartments or structures upon various stimuli. Unfortunately, however, isolation and expansion of single cells characterized by such easily-observable features is technically challenging, and indeed has not been accomplished to date.

**eLife digest** When scientists use microscopes to look at cells, they often want to then isolate certain cells based on how these look like. For example, researchers may want to select cells with specific shapes, movements or division rates, because these visual clues give important information about how the cells may be behaving in the body. However, it remains difficult to precisely pick a few live cells within a bigger sample.

To address this problem, Binan et al. created a new approach, called single cell magneto-optical capture (scMOCa), to set aside specific cells within a larger population. The technique uses the lasers present on confocal microscopes to attach tiny metallic beads to the surface of chosen cell. Then, a magnetic field is applied to gently pull the cell to a new location. The method is cheap – it relies on commonly available research tools – and it works on a broad variety of cells. In the future, scMOCa could be used to capture and then grow cells that can only be recognized by how they look or behave, which will help to study them in greater details.

DOI: https://doi.org/10.7554/eLife.45239.002

We recently developed a method termed Cell Labeling via Photobleaching (CLaP) (*Binan et al., 2016*) allowing the arbitrary tagging of individual cells among a heterogeneous population within a microscopy field. This is accomplished by crosslinking biotin molecules to their plasma membranes with the lasers of a confocal microscope, followed by use of fluorescent streptavidin conjugates to reveal the marked cells. In this manner, the same instrument used for imaging can also be adapted to label particular cells based on any visible trait that distinguishes them from the ensemble. Importantly, previous knowledge of surface markers or transfection of reporter genes are not required. Tags can be added with single-cell precision and the incorporated label displays convenient tracking properties to monitor location and movement. The mark is stable, non-toxic, retained in cells for several days, and moreover, does not engender detectable changes in cell morphology, viability, or proliferative capacity. Moreover, gene expression profiling indicated no major changes associated with the procedure (*Binan et al., 2016*). Nevertheless, a technology for the efficient isolation and expansion of CLaP-tagged cells is still lacking.

The fact that cell populations are often highly heterogeneous underscores the need for new approaches to capture and clonally expand individual cells of interest for further characterization. However, as mentioned above, current sorting techniques cannot efficiently isolate such rare cells (*Pappas and Wang, 2007*); indeed, classical protocols like Fluorescence and Magnetic Activated Cell Sorting (FACS and MACS) are typically optimized for high throughput at the expense of capture efficiency and specificity, and require large numbers of cells (*Pappas and Wang, 2007*). Small cell populations representing $10^{-3}$ of the total, which have been defined as rare, or ultrarare in the case of $10^{-5}$, can only be effectively captured and purified with repeated cycles of sorting and cell expansion protocols (*Pappas and Wang, 2007*). Starting with rare and hence precious cell populations, highly conservative gating strategies are needed, which can at best achieve approximately 45% purity (*Kuka, 2013*; *Shields et al., 2015*). Time-consuming manipulations, cost, hardware footprint, and handling complexity (*Takahashi and Okada, 1970*) make approaches based on microfluidics ill-suited for capturing small numbers of cells, which are often masked within tens of thousands.

Here, we report a novel technology, termed Single-Cell Magneto-Optical Capture (scMOCa), for isolating cells based purely on visual traits from within large heterogenous populations. After tethering biotin moieties to their membranes, cells of interest are targeted with streptavidin-coated ferromagnetic beads and captured with high efficiency using a simple magnet. The procedure is fast, uses low-cost commercially available reagents and only requires access to a standard confocal microscope. As proof-of-principle for the utility and power of this novel approach, we used scMOCa to i): capture and expand individual cells that differ in their capacity to resolve ionizing radiation (IR)-induced foci of the DNA repair protein 53BP1, ii) purify rare multinucleated cells, and iii) isolate cells that differentiated into adipocytes and accumulated lipid vesicles. Overall, the ease of use and affordability of our method is expected to facilitate the characterization of phenotypes of interest occurring in a small fraction of cell populations.

## Results

### scMOCa: efficient magnetic sorting of cells using ferromagnetic streptavidin-coated beads

#### Cell membrane biotinylation and ferromagnetic functionalization

We set out to evaluate whether individual cells illuminated with a low-power laser can be labeled with ferromagnetic beads, thereby facilitating their purification and clonal expansion. Adherent cells were incubated in medium supplemented with biotin-4-fluorescein (B4F), and a small area inside the cells of interest was illuminated with a 473 nm excitation laser at low power (<100 μW) for 2 s using a confocal microscope. This operation effectively crosslinks biotin molecules to plasma membranes and was repeated for all targeted cells. After washing, streptavidin-coated ferromagnetic beads were added to the medium, and then allowed to settle and attach specifically to illuminated cells (*Figure 1A*).

The high strength of the biotin streptavidin bond ($K_d = 10^{-15}$M) allows stringent rinsing and efficient removal of unbound magnetic beads, which is key to obtaining specific tagging allowed by the accurate laser pointing (*Figure 1B*). Depending on their size, beads may later be internalized (nanometer-size beads), or retained at the cell surface and shared between daughter cells after mitosis (micron-size beads). If needed, special beads, which integrate a DNA spacer between the streptavidin and their magnetic core to allow enzymatic cleavage, are commercially available (*Figure 1—figure supplement 1*). This permits detachment from cells in cases where beads can compromise downstream experiments, for example analysis of migration, or single-cell RNA sequencing.

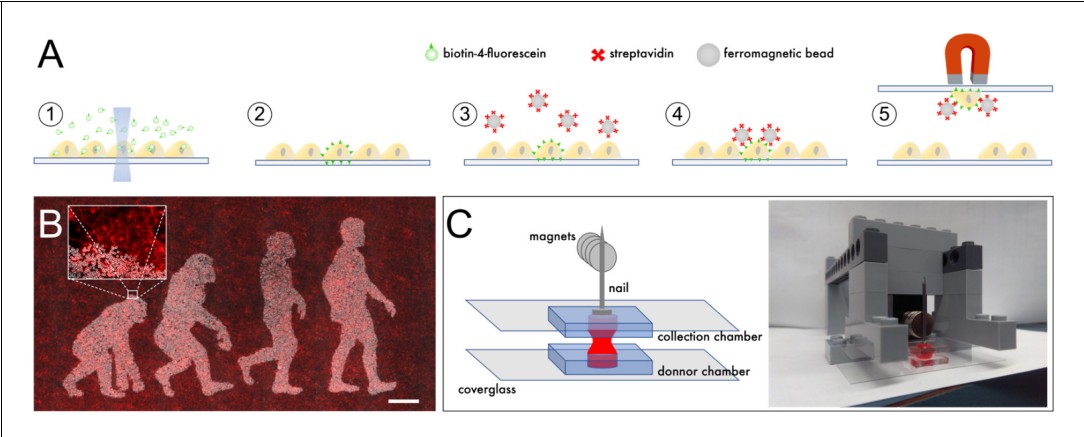

**Figure 1.** Outline of scMOCa. (**A**) Biotin-4-fluorescein is crosslinked to cell membranes with a laser. Biotin-tagged cells are labeled with streptavidin-coated ferromagnetic beads and captured with a magnet. (**B**) Example of a confluent U2OS cell culture where only cells illuminated with the lasers of a confocal microscope are densely decorated with magnetic particles. Beads appear in white, and all cellular membranes in red, tagged with WGA-Alexa647. Scale bar: 500 μm. (**C**) Schematic illustrating the simple tools needed to implement the protocol. Two small cell culture chambers cast in silicone and adhered to coverglasses are positioned one on top of the other. Cells in the bottom chamber are attracted to the top collection chamber by a magnetic field. A nail is placed above the collection chamber to guide the field generated by magnets to the donor chamber in which the cell suspension is kept. The collection chamber is held between two Lego bricks, filled with a solution of Trypsin (held in place by surface tension), and then slowly approached 6 mm above the bottom chamber, at which point the two drops merge.
DOI: https://doi.org/10.7554/eLife.45239.003

The following figure supplements are available for figure 1:

**Figure supplement 1.** Dettachment of magnetic beads.
DOI: https://doi.org/10.7554/eLife.45239.004

**Figure supplement 2.** Step by step protocol to tag and isolate cells using scMOCa.
DOI: https://doi.org/10.7554/eLife.45239.005

**Figure supplement 3.** Instructions to create a simple platform to hold both chambers, the magnets and the nail.
DOI: https://doi.org/10.7554/eLife.45239.006

**Figure supplement 4.** Three magnets were inserted inside a hollow Lego brick to magnetically hold the rest of the pile in position for sorting.
DOI: https://doi.org/10.7554/eLife.45239.007

## Rare cells can be sorted and expanded with high efficiency and specificity

We used trypsin to detach cells from the substrate before subjecting the entire population to a magnetic field that attracts labeled (positive) cells upwards to a collection chamber, while non-labeled (negative) cells remain in the original chamber. Specifically, two home-made chambers cast with silicone were filled with cell culture medium and positioned one on top of the other (*Figure 1C*). The top (receiving) chamber is also filled with trypsin and slowly brought together with the bottom chamber until both liquid drops merge. On top of the receiving chamber, a nail is placed to guide the magnetic field generated by a pile of 10 N35 magnets, each generating a 1.18 Gauss magnetic field at its surface (*Figure 1C*). Importantly, the nail must have high iron-alloy content for strong ferromagnetism. Only positive cells coated with ferromagnetic beads are pulled upwards to the top chamber, whereas negative cells are held down by gravity.

Magnets only attract positive cells with beads from the bottom well to the top well, regardless of the total number of cells in the sample. Repetition of the magnetic capture up to four times yields optimal selectivity: the collection (top) chamber can be simply flipped to replace the original donor chamber, while a new clean collection chamber is placed on top. The entire procedure takes only a few minutes and a detailed protocol is provided in Materials and methods and *Figure 1—figure supplement 2*. We note that a number of experimental parameters from this protocol need to be fine-tuned for specific cell types which exhibit different binding strengths and adhesion kinetics. In particular, the duration of the trypsin incubation, the number of times the capture is repeated, the time of exposure to the magnetic field, and the concentration of beads must be experimentally optimised.

Chamber dimensions can be critical for effective sorting, as their diameter (5 mm) and thickness (2 mm) determine the surface tension that holds liquid in the collection chamber and prevents it from falling. Furthermore, turbulence and movement must be avoided to prevent negative cells from reaching the collection chamber when both chambers are pulled apart. The distance that separates the two chambers while cells are being magnetically transferred must be maintained at approximately 6 mm such that gravity attracts negative cells as far away as possible from the collection chamber. The more distant the chambers are, the stronger the magnetic field must be to attract positive cells into the collection chamber; however, this could in turn affect the viability of transferred cells subjected to high pressure from beads pushing towards their cytoplasm.

We quantified the capacity of scMOCa to tag and isolate single cells from large populations. For this, we illuminated individual cells from chambers where approximately 50,000 cells had been seeded the day before and assessed capture efficiency. *Figure 2* shows examples where one or five cells were successfully sorted. Cells were non-specifically stained with WGA-Alexa-555 to facilitate detection and images were obtained before (*Figure 2*, left panels) and immediately after sorting (*Figure 2*, right panels). The right panels of *Figure 2* display both captured cells (visible in red) as well as unbound beads often aligned with the magnetic lines of force emanating from the head of the nail.

We have repeated these experiments and obtained similar results using both glass and Aclar (plastic) substrates, which vary significantly in their ability to promote cell adhesion. In every experiment, a given number of fluorescently labeled U2OS cells (1 to 50) were illuminated with a laser, sorted, and the receiving chamber examined to count captured cells. Cells in the receiving chamber with no visible beads attached to their membrane were considered as negative captured cells. *Figure 3A* demonstrates the high capture efficiency and selectivity of scMOCa, where blue dots correspond to experiments performed on Aclar substrates (higher cell adhesion) and red dots to glass (lower cell adhesion). Out of 23 experiments, starting from samples of 50,000 cells, the largest deviation from perfect recovery corresponds to one test where only 3, instead of 5 positive cells, were captured (two positive cells lost).

To further demonstrate the high specificity of our capture technique, that is to determine the ratio of false positive cells to the total number of chosen cells, 50 000 cells originating from two different species were co-cultured: MDCK (dog kidney cells) and IMCD (mouse kidney cells) at a 1:1 ratio. IMCD cells were incubated in WGA-Alexa 555 prior to mixing, to add a species-specific fluorescent marker. After 1 day in co-culture, the sample was brought to the microscope where 10 (nonfluorescent) MDCK cells were illuminated. We sorted the cells using scMOCa and performed PCR with primers specific for the cytochrome C gene from both dog and mouse. The results show that

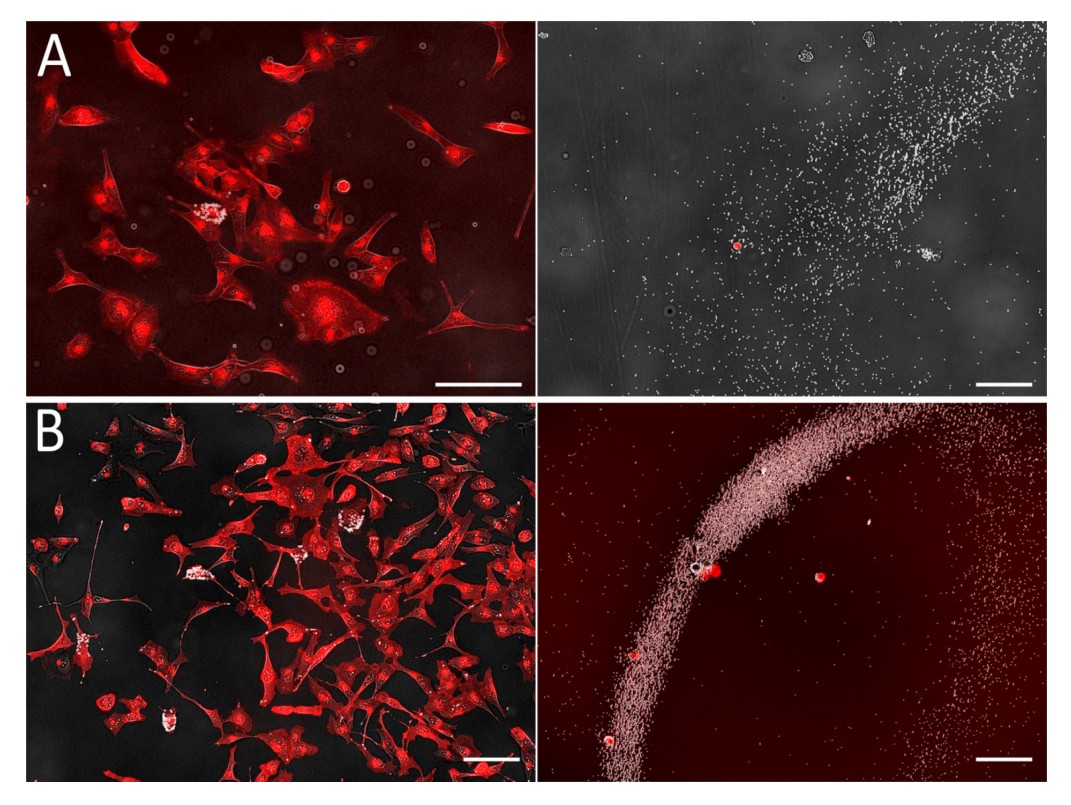

**Figure 2.** Images of cells functionalized with magnetic beads before (left, original chamber) and after (right, collection chamber) sorting. Beads appear in white (transmission image), and plasma membranes, tagged with WGA-Alexa555, in red (fluorescence image). Experiments were performed by tagging and sorting one cell (**A**) or five cells (**B**). In each case, it is apparent that all selected cells (left) are efficiently extracted (right) without contamination as the number of cells on the images on the right corresponds to the number of cells tagged. Tagged cells are easily recognized as they are covered with beads in both images. Scale bars: 50 μm.

DOI: https://doi.org/10.7554/eLife.45239.008

both cell types were present in the original mix, but only dog DNA was detected after magnetic sorting (*Figure 3B*). We also show by qPCR that these samples respectively contain an amount of DNA that corresponds to 10 and 9 dog cells, whereas mouse DNA is essentially undetectable (*Figure 3C*). We also note that since we amplified a mitochondrial gene present at hundreds of gene copies per cell, one negative cell or even a DNA dilution corresponding to less than one cell is expected to be detectable (DNA dilutions corresponding to less than one cell give readily detectable signals; see calibration curves in *Figure 3—figure supplement 1*). These experiments demonstrate that scMOCa isolates individual cells with high specificity. Indeed within a heterogeneous population, that is starting with a ratio 1:10,000 (positive: negative cells) in the source chamber, the method yields pure samples in the collection chamber. Our examples represent a five-orders-of-magnitude enrichment, as pure samples originating from a rare cell population (0.02% of the total) can be generated.

As a comparison to other capture methods based on magnetic fields, we prepared samples in which we sought to isolate 30 U2OS cells arbitrarily tagged amongst 30,000 by using commercially available separation columns (MACS, Miltenyi Biotec). These columns are optimized for high-throughput enrichment of large samples and are not designed for rare cells. In three independent experiments, we could isolate 5.3 ± 1.5 positive cells on average, while also capturing 17.6 ± 7.3 negative cells. This represents a population in which approximately 75% of the captured cells are contaminating false-positive cells with no beads attached, while scMOCa generates pure samples (*Figure 3A*). These results underscore the importance of the design of the home-made chambers and capture protocol, which prevents turbulent movement of cells.

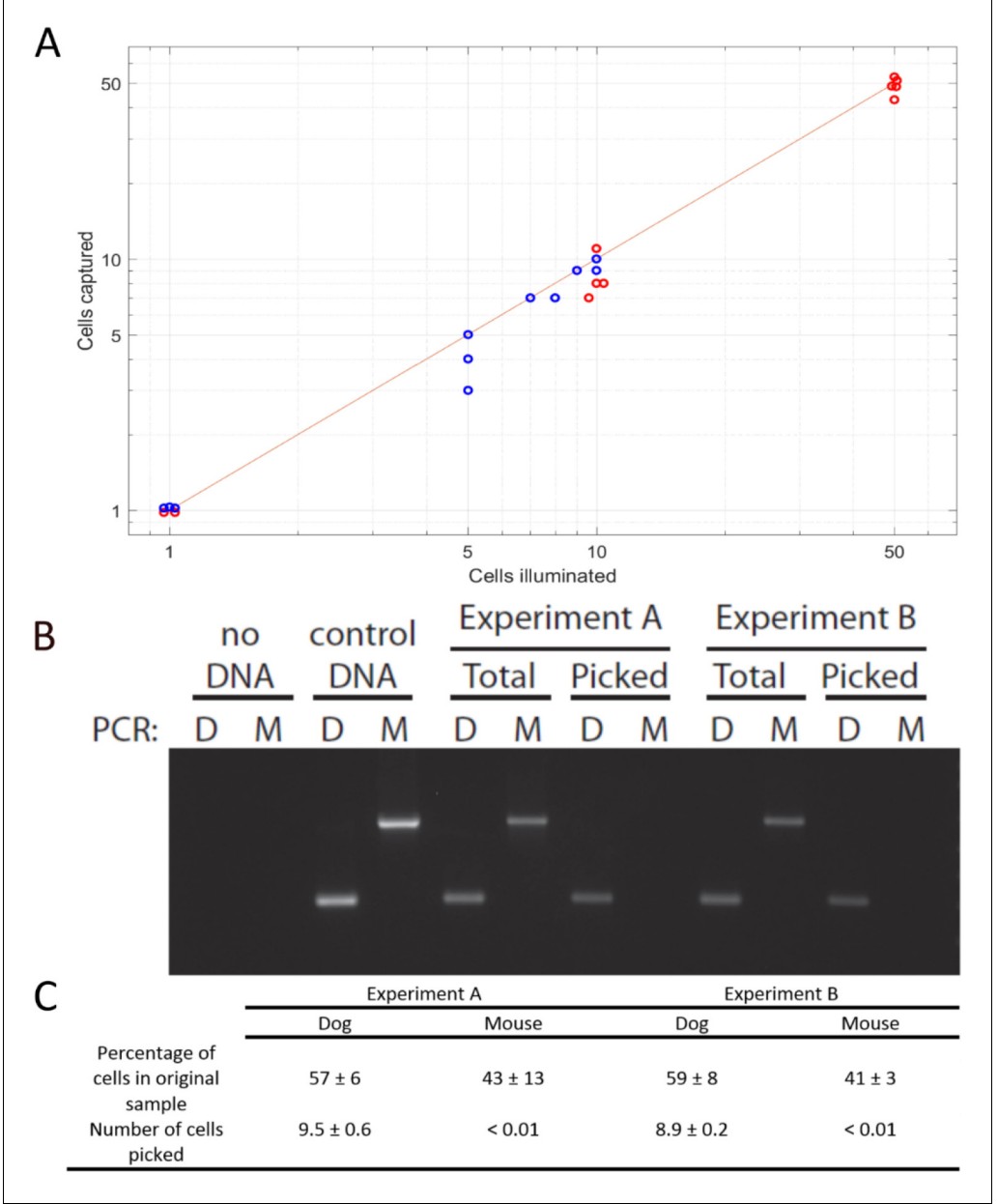

**Figure 3.** Capture efficiency and specificity. (**A**) Capture efficiency for 1, 5, 10, and 50 selected cells for a total of 27 experiments. Red dots represent experiments performed with glass as a cell culture substrate and blue dots correspond to experiments using Aclar as a substrate. The horizontal axis represents the number of target cells, considered as the number of cells illuminated with the laser. Ordinate axis shows the number of cells detected on the collection chamber after capture, and the line corresponds to 100% success rate. A linear fit of the data yielded a slope of 0.99, demonstrating that scMOCa is highly efficient in retrieving all target cells, after testing 1 to 50 cells. (**B, C**) Mouse (fluorescent) and dog (non-fluorescent) cell lines were co-cultured and only dog cells were illuminated and captured. PCR on a mitochondrial gene shows that all extracted cells form a pure sample and are exclusively dog cells. Table C shows the number of cells detected in each condition in three repeats of the experiment. These numbers are calculated from the amount of detected DNA normalized to the expected amount in one cell. A and B are independent experiments in which two different dishes were prepared, tagged and sorted prior to PCR.

DOI: https://doi.org/10.7554/eLife.45239.009

The following figure supplement is available for figure 3:

**Figure supplement 1.** Calibration curves used to calculate the number of cells from qPCR product.

DOI: https://doi.org/10.7554/eLife.45239.010

Cells can be placed back in culture and expanded after sorting. Immediately after capture cells are round (as expected after trypsin treatment), but after one day in culture they display normal elongated shapes (*Figure 4*). Upon proliferation the number of cells with beads attached is reduced exponentially as cells divide (*Figure 4*, right panels). In addition to immortalized cell lines, we have tested and successfully sorted three different types of primary cells: human umbilical vein endothelial cells (HUVECs), human lung fibroblasts, and mice dorsal root ganglion (DRG) neurons dissected and plated 24 hr before the assay. We specifically chose primary cells as these are known to be more fragile during manipulation than cell lines. Importantly, HUVECs and lung fibroblasts proliferated normally for several days and primary DRG neurons actively extended cellular processes, as shown in *Figure 4*. Finally, we tested mouse embryonic stem cells which, after capture and replating, displayed similar growth and morphological features relative to the original population. Indeed, cells sorted using gelatin-coated plastic chambers migrated and regrouped into small colonies which proliferated normally during 10 days. Sorted cells formed small poorly adherent spherical structures (*Figure 4D*) which is expected from embryonic stem cells as they are known to spontaneously form embryonic bodies in culture. Upon addition of 1 uM retinoic acid and removal of the leukemia inhibitory factor (LIF) from their medium, they started differentiating during five additional days (*Figure 4D*, right panel) and became more adherent cells spread on the culture substrate.

High plating efficiency is important when only one sample with very few cells needs to be expanded. Therefore, chamber culture conditions must be optimized for low cell numbers. Cell viability and proliferative potential can be improved by the use of conditioned medium (*Huang et al., 1990*; *Housden et al., 2015*; *Yamamoto et al., 2000*), that is, medium collected from an exponentially growing cell culture and passed through a 0.2 µm filter. This is attributed to secreted factors that in turn facilitate cell growth at very low density (*Huang et al., 1990*; *Yamamoto et al., 2000*). The top collection chamber can be coated with collagen to further improve cell attachment and viability (*Fradet-Turcotte et al., 2013*).

## Cells can be captured based on their ability to resolve ionizing radiation-induced DNA damage foci

To demonstrate the utility of scMOCa, we sought to isolate and expand cell populations based on their ability to resolve ionizing radiation (IR)-induced 53BP1 DNA damage foci, a well-characterized indicator of DNA double strand break (DSB) repair capacity (*Asaithamby and Chen, 2009*). For this, we used U2OS osteosarcoma cells harboring a construct permitting doxycycline-inducible expression of 53BP1 fused to Green Fluorescent Protein (GFP). 53BP1 is directly involved in DSB repair and is rapidly recruited to DSB sites where it forms foci that can be readily detected by fluorescence microscopy in live-cells (*Mirzayans et al., 2018*) when fused with GFP. Foci of 53BP1 are resolved gradually as cells repair DSB, and within approximately 3 hr post-irradiation with 0.5 Gy most are expected to disappear (*Mirzayans et al., 2018*).

We exposed cells to 0.5 Gy of IR and imaged GFP-53BP1 foci. We first characterized focus formation and resolution by measuring the average number of foci before and after IR in 500 cells. At 45 min post-irradiation an average of $10.2 \pm 2.5$ (mean $\pm$ standard deviation) foci per cell was detected. At 2 hr post-irradiation, a second set of images was acquired, and the average number of foci was reduced to $7.6 \pm 2.3$. Since on average cells resolved approximately 25% of their foci within 2 hr, we defined cells in which more than 85% of foci have disappeared after 2 hr as 'fast resolving'. Such fast resolving cells, represented approximately 1% of the population. In all following experiments, we compared both sets of images to search for fast-resolving cells (two such cells are shown in *Figure 5A*) and used scMOCa to tag, capture and expand them.

We emphasize that FACS or similar approaches are not suitable for sorting based on focus resolution, even if the fraction of target cells was relatively large, as the overall fluorescence signal originating from cell nuclei does not reflect the local distribution of protein. Indeed, we observed no change in global protein abundance or average intensity of GFP-53BP1 upon focus resolution: the average intensity of nuclei showed no correlation with the number of 53BP1 foci (Pearson coefficient of $-0.15$). Because we used very stringent selection criteria for focus resolution, we tagged only 5 and 3 'fast-resolving' cells in two independent experiments, which were subsequently isolated using scMOCa, pooled and expanded to generate Populations #1 and #2.

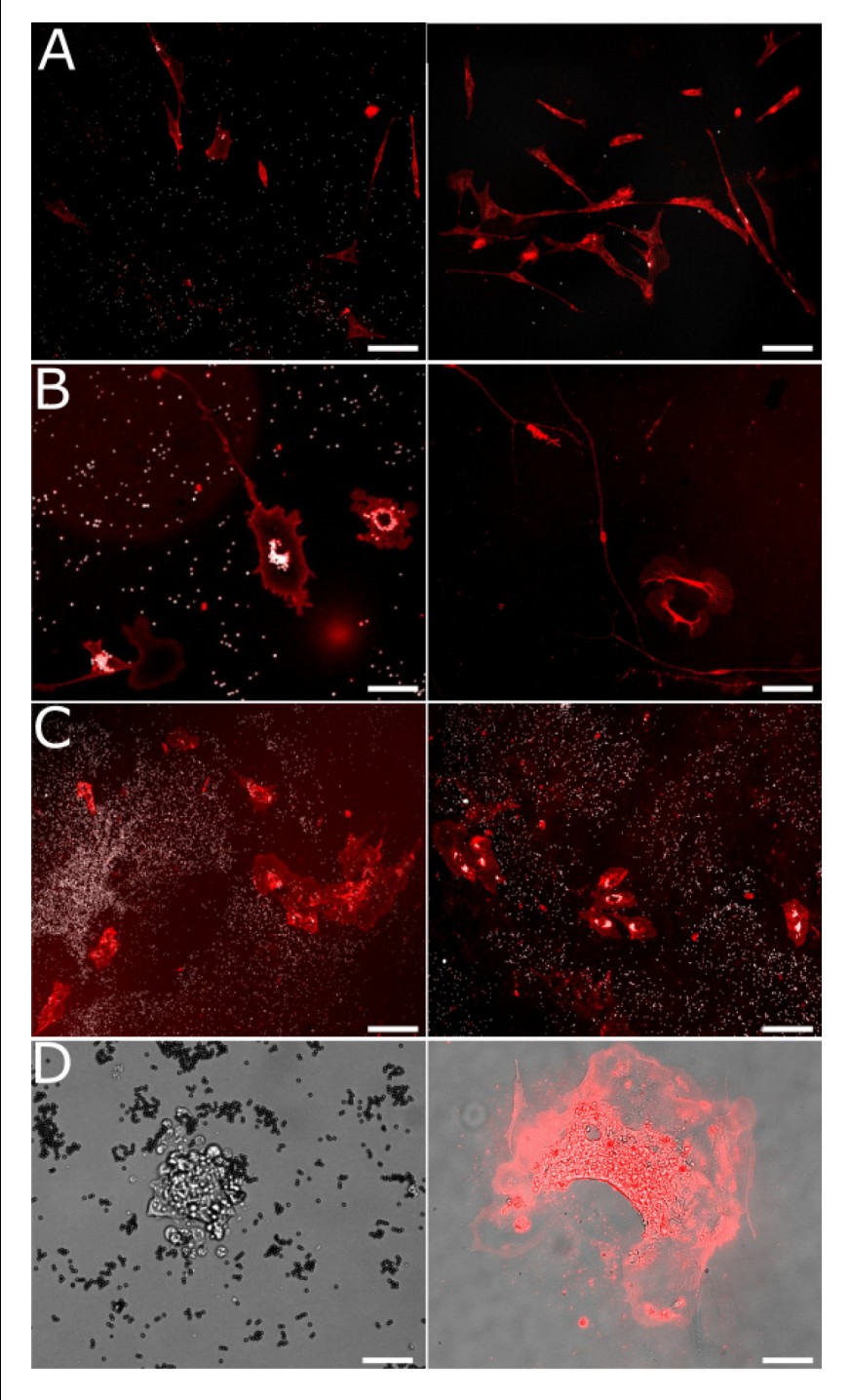

**Figure 4.** Cells remain viable and proliferate after capture. Images showing scMOCa-captured cells stained with WGA-Alexa-647. (**A**) LF-1 fibroblasts 1 (left) and 4 (right) days after sorting. Scale bar 80 μm. (**B**) Primary DRG neurons 2 (left) and 4 (right) days after sorting. Scale bars: 25 μm (left) and 80 μm (right) (**C**) HUVECs 3 (left) and 6 (right) days after sorting. Scale bar: 80 μm. (**D**) Mouse embryonic stem cells 7 days after sorting (left) and 5 days after starting differentiation (15 days after sorting) (right). Prior to differentiation, only a bright-field image is shown to preserve cell viability. After differentiation, we stained cells with WGA-Alexa647, and merged the image with a bright-field photo to increase contrast and better see cellular extensions. Scale bar: 40 μm.
DOI: https://doi.org/10.7554/eLife.45239.011

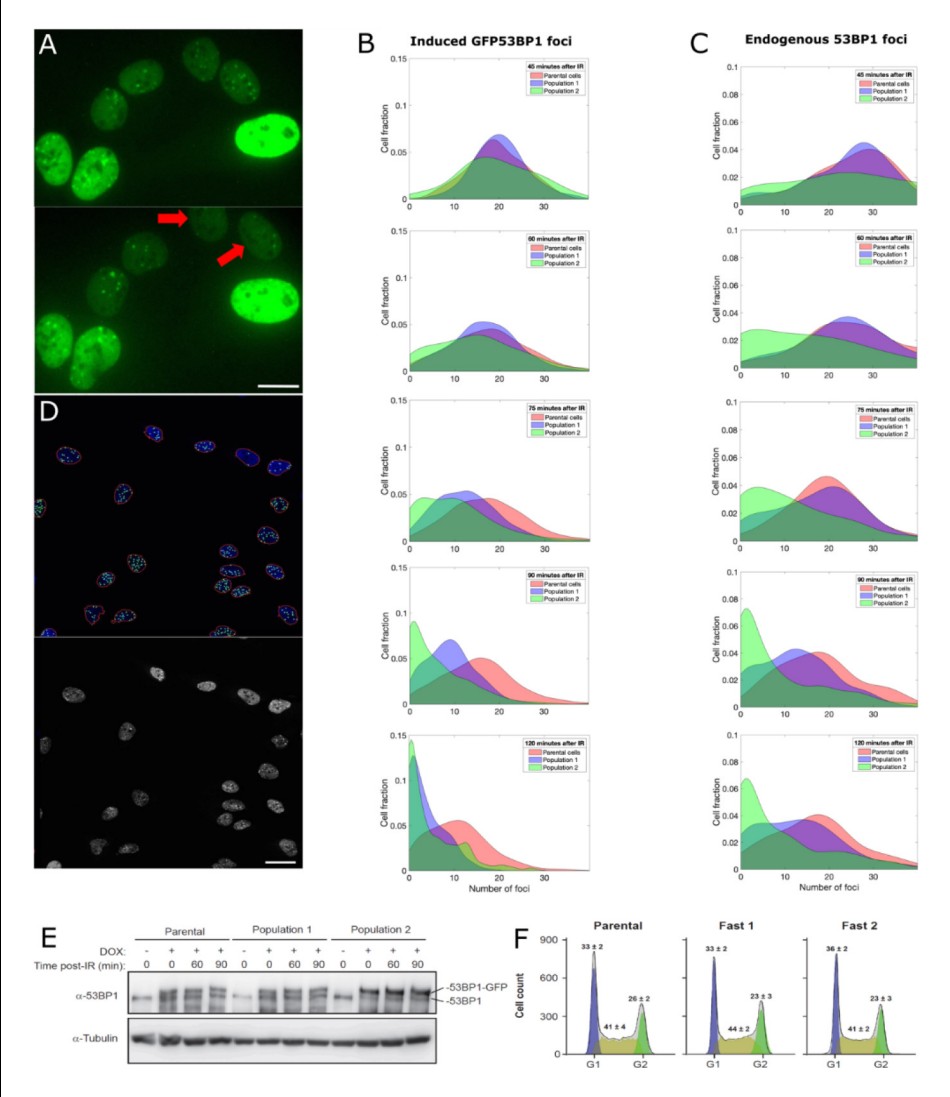

**Figure 5.** Capture and expansion of individual cells that differ in their capacity to resolve ionizing radiation-induced 53BP1 foci. (**A**) Nuclei from irradiated cells 40 min (top) and 90 min (bottom) post-irradiation. Two cells (red arrows) resolved 53BP1 foci more rapidly and were selected for capture. Scale bar: 14 μm. (**B, C**) Smoothed normalized histograms showing the fraction of cells detected as a function of the number of induced GFP-53BP1 (**B**) or endogenous 53BP1 (**C**) foci for five time points. Sorted Populations #1 and #2 resolve foci faster than their parental counterpart as illustrated by the more rapid shift toward the left (zero foci per cell) observed for these two populations. (**D**) Illustration of automatic nuclei segmentation and detection of foci (top) and source image (bottom). Objects detected as nuclei are circled in red, segmented foci appear as green circles. Scale bar 25 μm. (**E**) Immunoblot showing the amount of 53BP1 at 0, 60, 90 min post-irradiation in doxycycline induced cells (+) and non-induced cells (-). 53BP1 levels are not altered in Populations #1 or #2 compared to the parental cells. (**F**) Cell cycle profiles of U2OS GFP-53BP1 parental cell lines and two extracted populations. Cultures were induced with Dox for 48 hr and cell cycle was analyzed by DNA content flow cytometry (see Material and methods). Values represent the means ± SEM of three independent experiments. All focus quantification graphs represent the average of 3 experiments, where in each case at least 200 cells were scored.

DOI: https://doi.org/10.7554/eLife.45239.012

## The ability to quickly resolve 53BP1 foci is transmitted from parental to daughter cells

We next compared the kinetics 53BP1 focus resolution in Populations #1 and #2 vs. the parental cell population. The resolution of foci was quantified using (i) live-cell imaging of GFP-53BP1 (*Figure 5B*)

and also (ii) following immunostaining with anti-53BP1 antibody (when GFP-53BP1 expression was not induced) to evaluate focus formation involving the endogenous untagged protein (*Figure 5C*). Images were acquired at 45, 60, 75, 90 and 120 min post-irradiation with 1Gy for the two populations and the distribution of DNA foci per cell compared with that of the parental cell line. We used Matlab to program a fully automated algorithm for focus quantification (*Figure 5D*) and analyzed approximately 1800 cells per time-point. This allowed the unbiased evaluation of large datasets as *Figure 5B and C* taken together represent the behavior of more than 21,000 cells.

*Figure 5B and C* shows normalized histograms (probability density functions) of the number of foci per cell at each time-point. Importantly, all three populations exhibited similar numbers of foci per cell 45 min after irradiation, indicating that the initial formation of 53BP1 foci is comparable between all cell populations. However, we found that the progeny of captured cells (Populations #1 and #2) retained the original visually detected phenotype of fast focus resolution. These cells resolved foci at least 1.5 times more rapidly than parental counterparts, as the median number of GFP-53BP1 foci per cell 60 min post-IR for Populations#1 and #2 (17 and 15 respectively) is equal to the median number of foci that parental cells exhibit at 90 min post-IR. After 75 min, these numbers of foci are already statistically different (p-values from student T-tests comparing the parental cells to Populations #1 and #2 are respectively $10^{-75}$ and $10^{-39}$). Such differences in focus resolution dynamics is particularly striking in cells for which the expression of GFP-53BP1 is induced (*Figure 5B*) but is clearly observable as well using immunofluorescence of the endogenous protein in non-induced fixed cells (*Figure 5C*).

To rule out the possibility that resolution of 53BP1 foci might be due to increased degradation upon IR or to globally decreased levels of the protein, we monitored 53BP1 levels by immunoblotting at different time points post-IR. No changes in the levels of either endogenous 53BP1 or GFP-tagged version was observed (*Figure 5E*). Finally, FACS analysis shows that all populations exhibit similar ratios of cells in each cell cycle phase (*Figure 5F*). Therefore, the observed focus resolution differences between populations is unlikely to be attributable to cell cycle-related effects.

## Cells can be purified based on morphology

We next sought to illustrate of the utility of scMOCa to capture cells based on their morphology, which have so far proven challenging to sort using currently available technologies. For example, multinucleated cells constitute a rare subpopulation (*Mirzayans et al., 2017*; *Coward and Harding, 2014*) that does not express specific markers and cannot be differentiated from mononucleated polyploid cells using DNA-specific stains in a FACS experiment. However, multinucleated cells can be easily identified visually even without DNA staining. In the context of cancer, such cells have been (i) described as generally being more aggressive and metastatic than mononucleated counterparts, and (ii) proposed to be prone to acquisition of drug resistance and cancer relapse (*Mirzayans et al., 2017*; *Mittal et al., 2017*; *Weihua et al., 2011*; *Green and Meuth, 1974*). Moreover, even though multinucleated cells do not undergo classical cytokinesis, they can generate mononucleated progeny by budding (*Mirzayans et al., 2017*; *Weihua et al., 2011*) and influence neighboring cells by secreting factors that promote stemness, as well as by transmitting sub-genomes (*Mirzayans et al., 2017*).

Multinucleated cells were isolated using scMOCa and kept in culture for 4 days to evaluate their viability and metabolic activity (*Figure 6*). We used WGA-alexa647 to stain plasma membranes, and Hoechst for the nuclei (*Figure 6*) and Mitotracker green FM to tag polarized mitochondrial membranes, indicating that scMOCa preserves the viability of isolated cells (see *Figure 6—figure supplement 1*).

As another example of a visual phenotype that can be sorted using scMOCA, we evaluated the differentiation of 3T3 cells into adipocytes. These cells are amongst the most common models to study metabolic disorders, for example, obesity (*Armani et al., 2010*; *Majka et al., 2014*). When cultured for 2 days in medium containing dexamethasone, insulin and isobutylmethylxanthin (IBMX), an inhibitor of cyclic nucleotide phosphodiesterases, and 3 days in medium containing insulin, a fraction of 3T3 cells differentiate and lipid vesicles accumulate in their cytoplasm. In order to obtain pure adipocyte cultures, flow cytometry sorting based on granularity requires several steps to select cells of interest and then remove false positives, such as debris and cell aggregates (*Nagrath et al., 2007*), whereas scMOCA may provide a much simpler approach to isolated live adipocytes, especially when these are present in very low abundance. We used scMOCA to capture differentiated adipocytes

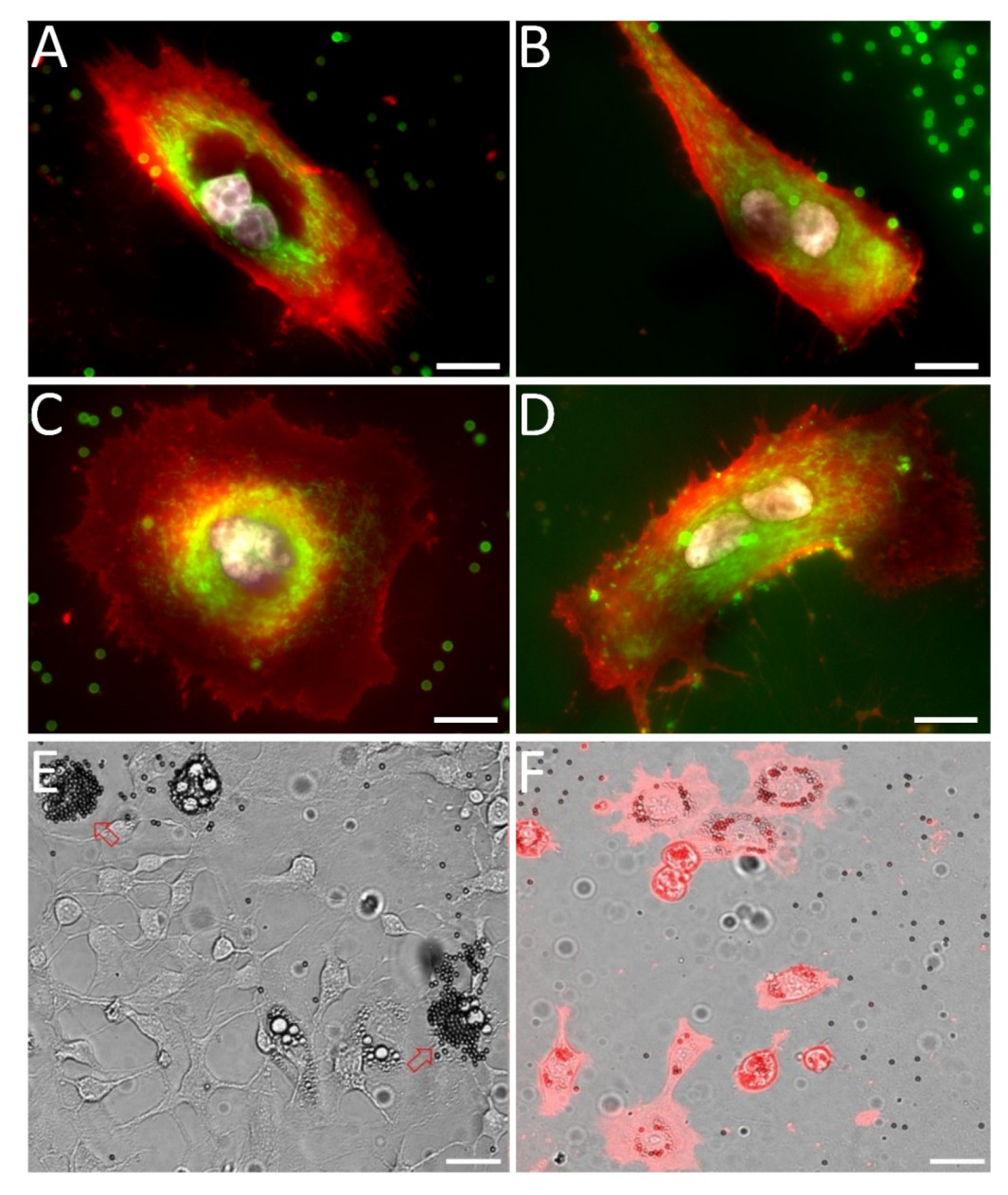

**Figure 6.** Examples of sorted multinucleated H226 cells (**A**, **C**) 1 and 2 (**B**, **D**) days after scMOCa. Active mitochondria (Mitotracker) appear in green, plasma membrane (WGA-Alexa 647) in red, an nuclei (Hoechst) in white. Scale bar: 15 μm. (**E**) 3T3 cell population partially differentiated into adipocytes. Two cells (pointed by arrows) have been tagged with magnetic beads. Three other cells are also differentiated in adipocytes in this field of view but were not selected. (**F**) Cells were captured using scMOCA and kept in culture for 6 days before imaging. Cells were stained with WGA-Alexa 647 to highlight membranes. Small black circles are magnetic beads while lipid vesicles appear as small clear circles. Scale bar: 30 μm.

DOI: https://doi.org/10.7554/eLife.45239.013

The following figure supplement is available for figure 6:

**Figure supplement 1.** Two examples of cells that were killed with sodium azide then stained with mitotracker green and Hoechst.

DOI: https://doi.org/10.7554/eLife.45239.014

and then kept them in culture for a week (*Figure 6*). Sorted cells remained viable and maintained their ability to store lipids in vesicles that appear as clear spheres on *Figure 6*, while the magnetic beads that remained attached to cells membranes appear as dark spheres.

## Discussion

To the best of our knowledge, scMOCa is the only technology that permits isolation, and subsequent clonal expansion, of extremely small numbers of cells from relatively large heterogeneous populations based solely on visual criteria. scMOCa is highly efficient, as the fraction of tagged cells collected in the top chamber exhibits minimal capture losses and high specificity. Rare false positive cells, presumably attached by cell junctions to true positive cells, can be eliminated by repeating the sorting procedure to reach 100% purity. The most widely used cell sorting technique, FACS, is not optimized for sorting rare cells. Adaptations needed for capturing cell populations representing <1% of the sample with high specificity make FACS experiments cumbersome and inefficient. Moreover, repetition of flow cytometry sorting to obtain pure samples of a given cell type imposes can only be performed with robust cell types due to reduced survival and proliferation capacity (*Pappas and Wang, 2007*). More refined procedures have been developed to sort rare cells via binding to microfluidic channels coated with antibodies against specific surface markers of interest (*Antfolk et al., 2017*). However, this requires high-affinity antibodies that are specific to the target cell types and leads to dilution of cells in laminar flows within microfluidics chips (*Moon et al., 2011*; *Kang et al., 2012*), which can become a drawback for downstream applications. Techniques based on magnetism display an improved capacity to isolate rare cells without dilution (*Tham et al., 2014*). Nevertheless, while the majority of protocols that use magnetic fields can capture cells of interest with efficiency near 90%, their specificity remains a major challenge, as published results vary between 10% and 80% purity for captured cells (*Miltenyi et al., 1990*), generally closer to 50% (*Zborowski and Chalmers, 2011*; *Pamme and Wilhelm, 2006*; *Radbruch et al., 1994*; *Khojah et al., 2017*). Finally, only a handful of approaches allow label-free cell sorting, where intrinsic physical properties, such as size (*Zhao et al., 2017*; *Monti et al., 2017*) or magnetic susceptibility (*Moon et al., 2011*; *Pamme and Wilhelm, 2006*; *Hosokawa et al., 2010*) differentiate the target population. Filtration, for example, relies on porous membranes to capture cells based on size and deformability (*Davis et al., 2006*; *Gascoyne et al., 2009*) and can achieve 80% efficiency. Dielectrophoresis exploits natural differences in dielectric properties of cell types for discrimination and circulates cells in microfluidics channels, deviating target cells within an electric field (*Hu et al., 2005*; *Landry et al., 2015*).

The application we introduced here is focused on magnetic separation, but the same concept of adding particles to individual live cells may open the door to novel strategies where other actionable properties can be exploited in a simple and straightforward manner. For example, fluorescence or electron density can be manipulated on single cells (*Binan et al., 2016*), and recent advances in cellular nanotechnologies such as scattering and plasmon resonance using gold nanoparticles, thermal capacity with nanoshells, or electrical properties using carbon nanotubes can now be modulated only on chosen cells using low-cost commercially available reagents.

ScMOCa presents critical advantages over more traditional sorting techniques. It allows isolation of live cells without previous knowledge of surface markers and can simply be based on morphological traits such as the presence of nuclear foci or lipid vesicles and the number nuclei. More importantly, it has the potential to sort based on time-dependent characteristics such as migration speed or foci resolution. In addition, because sorting is carried out in small chambers of similar size, there is no sample dilution. This prevents cells from sustaining strong shear stress upon passing through microfluidic tubing (*Miltenyi et al., 1990*), and allows their use in downstream applications such as cell culture, reinjection, or even lysis prior to transcriptomic or proteomic analysis. ScMOCa cross-links biotin to cell membrane and the strength of the ensuing biotin-streptavidin bond is extremely high ($K_d = 10^{-15}$M). In comparison, the bonds utilized in immunochemistry are much weaker, from $10^{-12}$ to $10^{-9}$ [39,40], which may cause tags to detach from cells because of shear stress within the microfluidics tubing (*Wooldridge et al., 2009*). Another example is provided by ligands targeting the major histocompatibility complex (MHC) on immune cells where binding strength is so weak that ligands usually need to be grouped in tetramers for increased strength (*Tsai et al., 2004*; *van der Toom et al., 2017*). Finally, while the precise mechanisms influencing 53BP1 focus resolution was

not investigated in our proof-of-principle experiments, our data demonstrates that markers used for identification need not be exposed on the membrane since the spatial distribution of fluorescent signal originating from the nucleus were used here as a reporters.

Simplicity is a key advantage of scMOCa, as it does not require highly specialized software, or hardware such as microfluidic chips. Indeed, a standard confocal microscope with no modification, simple handmade chambers and low-cost magnets are all that is needed to sort single cells of choice from among tens of thousands. The main limitation of scMOCa is that high throughput implementations would depend on efficient image processing tools for cell detection. While automated detection and tagging are possible on motorized microscopy systems, the duration of the procedure is roughly proportional to the number of target cells. Thus, even if laser illumination of a single cell typically requires one second, this might become a limitation for applications that deal with large cell numbers.

The capacity of scMOCa to isolate and profile individual cells within a large population based purely on visual phenotypes constitutes a powerful tool for understanding cellular heterogeneity. We envision that one potential application of high interest would combine scMOCa with single cell sequencing to characterize the molecular basis of differential metastatic potential among particular cells within a tumour (*Navin et al., 2011*; *Valastyan and Weinberg, 2011*; *Shapiro et al., 2013*; *Tirosh et al., 2016*; *Heitzer et al., 2013*; *Gierahn et al., 2017*). Indeed, scMOCa can easily be combined with currently available techniques that allow sequencing RNA from single cells captured in wells (*Brennecke et al., 2013*) and microfluidic chips (*Wu et al., 2014*; *Tan et al., 2017*). More generally, it is becoming increasingly obvious that the capacity to analyze rare cells in heterogeneous populations will be useful in designing personalized treatments for cancer (*Hood et al., 2004*; *Pugia et al., 2017*) as well as for inflammatory, autoimmune, and neurologic disorders (*Miltenyi et al., 1990*; *Weissleder, 2009*; *Hesketh et al., 2017*).

# Materials and methods

**Key resources table**

| Reagent type (species) or resource | Designation | Source or reference | Identifiers |
|---|---|---|---|
| Cell line (*Homo sapiens*) | U2OS | ATCC | RRID: CVCL_0042 |
| Cell line (*Canis familiaris*) | MDCK | ATCC | RRID: CVCL_0422 |
| Cell line (*Mus musculus*) | IMCD | ATCC | RRID: CVCL_0429 |
| Cell line (*Homo sapiens*) | h226 | ATCC | RRID: CVCL_1544 |
| Cell line (*Homo sapiens*) | LF-1 | Dr John Sedivy | RRID: CVCL_C120 |
| Cell line (*Homo sapiens*) | HUVECS | ATCC | TCC PCS-100–013 |
| Cell line (*Mus musculus*) | 3t3-L1 | ATCC | RRID: CVCL_0123 |
| Chemical compound, drug | IMBMX | Sigma-aldrich | cat #: I5879-100MG |
| Chemical compound, drug | Dexamethasone | Sigma-aldrich | cat #: D1756-25MG |
| Chemical compound, drug | Magnetic beads | Thermofisher | cat #: 65305 |

*Continued on next page*

*Continued*

| Reagent type (species) or resource | Designation | Source or reference | Identifiers |
|---|---|---|---|
| Chemical compound, drug | b4f | Sigma-aldrich | cat #:B9431-5MG |
| Commercial assay or kit | 2X SYBR Green Master Mix | Bimake | cat #: B21203 |
| Antibody | Rabbit anti-53BP1 | Santa-cruz | cat #: sc-22760 |
| Antibody | Rat anti-tubulin | Abcam | cat #: ab6161 |

## Cell culture

U2OS osteosarcoma cells, MDCK (dog) cells, and IMCD (mouse) cells were grown in DMEM/F12 medium supplemented with 10% FBS and antibiotics, all purchased from Thermofisher Scientific. One day prior to the experiment, cells were detached and seeded on either collagen-coated glass coverslips or circular pieces of Aclar (polychlorotrifluoroethylene) coated with collagen, onto which polydimethylsiloxane (PDMS) chambers had been placed (see below).

A U2OS cell line with inducible expression of GFP-tagged 53BP1 was constructed as previously described (*Al-Hakim et al., 2012*) using pcDNA5-FRT/TO-eGFP-53BP1 (*Fradet-Turcotte et al., 2013*) (Addgene plasmid #60813) and the U2OS Flip-In TREX host cell line (*Brown et al., 1997*) (both generous gifts from Dr. Daniel Durocher, University of Toronto). Cells were selected in medium supplemented with 200 μg/mL hygromycin and 5 μg/mL blasticidin. GFP-53BP1 expression was induced by addition of 5 μg/mL doxycycline for 48 hr.

H226 cells were grown in RPMI medium supplemented with 5% FBS and antibiotics (Thermofisher Scientific). Four days prior to the experiment, cells were exposed to 6 μg/mL cytochalasin B for 24 hr. Low-passage primary human lung fibroblasts (LF-1) were a kind gift from Dr John Sedivy (*Talbot et al., 2015*). Cells were grown in Eagle's MEM (Corning) containing 15% FBS, essential and nonessential amino acids, vitamins, L-glutamine, and antibiotics (Life Technologies). HUVECS were grown in Endogro TM (Millipore) supplemented with VEGF. Primary dorsal root ganglion (DRG) neurons were harvested from IsI-Gcamp6 x TRPV1-cre mice and cultured in plastic bottom dishes (as detailed elsewhere [*Bélanger et al., 2018*]) one day prior to the sorting.

## 3T3-L1 cell culture and adipogenic differentiation

Pre-adipocyte 3T3-L1 cells were grown in DMEM medium supplemented with 10% FBS (Gibco), 2 mM glutamine (Wisent) and 1% Penicillin/Streptomycin (Biobasic). For adipogenic differentiation of 3T3L1, the cells were plated at confluency and media was changed to induction media containing 10% FBS, 1% Penicillin/Streptomycin, 1 μM Dexamethasone, 1 μg/ml Insulin and 500 μM IBMX (Sigma). Two days post-induction, the medium was changed to maintenance media containing 10% FBS (Gibco), 1% Penicillin/Streptomycin (Biobasic), 1 μg/ml Insulin. After 3 days post-induction, 10,000 cells were plated on homemade chambers for sorting.

Mouse Embryonic Stem cell (mES) culture mES cells were grown in DMEM medium supplemented with 15% FBS (embryonic stem cell qualified, Wisent), 1 X non-essential amino acids (Sigma), 100 μM 2-Mercaptoethanol (Gibco), 1000 Units/mL Leukemia inhibitory factor (LIF, Stemcell), 2 mM glutamine (Wisent) and 1% Penicillin/Streptomycin (Biobasic) on 0.1% porcine gelatin-coated plastic dishes (Sigma). About 10,000 cells were plated for sorting as above.

## PDMS chambers

PDMS chambers were prepared by pouring a mix of resin and curing agent (10:1 ratio) in a petri dish to achieve a gel thickness of 2 mm. The dish was degassed overnight in a vacuum chamber and the resin allowed to polymerize at room temperature for 2 days. Square pieces were cut with a blade, circular wells of 5 mm diameter were made using a biopsy punch from Miltex (33-38) (see *Figure 1B and C*) and placed on either glass or Aclar coverslips (onto which PDMS naturally adheres).

## scMOCa protocol

Cells were incubated in regular medium with 40 µg/mL biotin-4-fluorescein (Sigma) on glass coverslips or Aclar substrates. A spot within each cell of interest was illuminated at 473 nm with the laser of a confocal microscope at 75 µW for 2 s with 10 × 0.4 NA objective. The sample was then thoroughly rinsed in PBS, and medium containing 8 µL of streptavidin-coated ferromagnetic beads of 2.8 µm in diameter (Thermofisher, 65305 and 11533D) was added. When beads were attached to a whole area rather than a single cell (*Figure 1B* and *Figure 1—figure supplement 1*) the sample was scanned with a 700 µW laser scanned at 0.2 mm/s with a 0.4 NA objective in a succession of lines 0.005 mm apart to form a pattern generated from a binary image.

Beads were pulled down in contact with the cells and re-suspended 3 times, attracted by a magnet placed alternatively below or above the sample. Cells were then rinsed thrice with PBS and a magnet was positioned above the sample to remove unbound beads. After this, very few beads remain in the dish (*Figure 1C*).

Cells are detached using 0.25% trypsin (Thermofisher, 25200072) for magnetic capture. The resulting cell suspension is then subjected to a magnetic field that attracts positive cells upwards to a collection chamber, while negative cells settle by gravity in the original chamber, regardless of the total number of cells in the sample.

More specifically, once the original PDMS culture chamber contains a suspension of individual cells in trypsin, a second identical PDMS culture chamber is placed on top of the first one as depicted in *Figure 2A*. The structure that holds the top chamber in place can be built with Lego bricks (*Figure 1—figure supplements 3* and *4*): the collection chamber is positioned between two Lego bricks that maintain it at 6 mm above the cells (*Figure 1C*). While magnetic attraction of tagged cells toward the collection chamber is quick, negative cells require 4 min to settle down to the original chamber before the top chamber is separated, flipped, and the magnets removed. This procedure needs to be performed slowly to minimize turbulence and to avoid capture of negative cells.

These manipulations are repeated three times to attain maximum specificity (*Figure 2C*). The collection chamber is always filled with trypsin solution to avoid rapid cell adhesion, and gentle up and down pipetting can be performed to prevent cell clumping. Only for the last capture is the collection chamber filled with medium in which the cells will be expanded. The entire procedure is summarized in *Figure 2C*.

Experimental conditions need to be fine-tuned for different cell types. The most important parameters that need to be optimised are surface coating of both donor and collection chambers, duration and number of repeats of the sorting steps. The collection chamber should provide optimal plating efficiency to maximize cell survival of very few cells while the donor chamber should allow strong adhesion of the cells to allow thorough rinsing of free magnetic beads. In our experience collagen coating provides strong cell attachment, but also generates extracellular fibers where beads and negative cells can be entangled and captured. Gelatin solves the issue of collagen fibers, but cell adhesion is slightly reduced, which may cause cell loss during rinsing. Uncoated substrates are an easy solution for cells like U2Os but many cell types including primary cells do not proliferate well on such surfaces. Plastic bottom chambers allow better cell adhesion and survival, but their reduced optical quality may hamper the precise observation of selection criteria. In this respect, Aclar possesses excellent optical properties and represents an excellent alternative. For most cell types, longer incubations (approximately 4 min) allow negative cells to settle down in the donor chamber, reducing the number of repeats required for optimal purity. On the contrary, experimentation with cells that adhere rapidly (e.g. MDA-MB-231), require the capture protocol to be performed as quickly as possible and more repeats may be needed.

In our hands, the best results were obtained using 10 magnets each generating a 1.2 Gauss magnetic field and 2 mm deep PDMS wells. In this condition, it is important that the distance between the bottom of each chamber is kept at 6 mm to allow the magnetic field to attract all tagged cells against gravity to the collection chamber while preventing the turbulence generated by the separation of the chambers to bring negative cells into the collection chamber. Increasing this distance requires a stronger magnetic field, which in turn reduces viability of captured cells. The diameter of the chambers should also be 5–6 mm, to ensure the necessary surface tension that allows merging and splitting the media in both donor and collections chambers.

## Cell sorting using commercial magnetic cell separation columns

30,000 U2Os cells were plated in our homemade chambers 1 day prior to sorting. On the day of the experiment, 30 cells were arbitrarily chosen and tagged in three independent experiments. We manually counted and verified that the right number of cells (30) were covered with magnetic beads in each dish. Commercial MACS columns were washed with PBS containing 0.5% BSA and 2 mM EDTA as indicated by the manufacturer. Cells were detached using 60 μL trypsin and then diluted in 500 μL of the same buffer and placed in the column in the magnets from Miltenyi Biotec. Columns were rinsed three time with buffer, then removed from magnets and washed with 5 mL buffer. Cells were then centrifuged, resuspended in 70 μL medium and placed in new homemade chambers for observation and counting under the microscope. Any cell that had visible magnetic beads on its membrane was considered as a positively selected cell, while cells free of beads were counted as negative cells.

## Identification and isolation of 'fast resolving' live cells

Forty-eight hours after induction of GFP-53BP1 with doxycycline, U2OS cells were irradiated with 0.5 Gy of IR. A first set of images was acquired with a 40X, 0.95 NA objective 45 min post irradiation, to detect focus formation.

Cells that displayed a > 85% reduction in the number of foci at the second time point (2 hr) were considered 'fast-resolving'. Biotin-4-fluorescein (0.04 mg/mL) was then added to the medium, and such cells were illuminated for 2 s through a 10 × 0.4 NA objective with 75 μW of laser intensity at 473 nm.

## Immunofluorescence and automated detection of nuclear GFP-53BP1 foci

Immunofluorescence was performed to evaluate levels of endogenous 53BP1 foci. Briefly, cells were rinsed with PBS, and fixed 15 min with 4% paraformaldehyde in PBS. Cells were then permeabilized for 10 min with 0.5% Triton X-100 in PBS, rinsed twice in PBS and twice in PBS + 0.05% Tween-20 and then blocked in PBS + 3% BSA and 0.05% Tween20. Rabbit anti-53BP1 antibody (Santa-Cruz) was diluted 1:500 and incubated on the cells for 3 hr. Cells were rinsed in PBS + 0.05% Tween-20 and incubated with Alexa-488 anti-rabbit for 1 hr, washed three additional times, and finally imaged for focus quantification.

An image processing pipeline was programmed to fully automate DNA focus detection as we have previously done (*Bélanger et al., 2016*; *Otsu, 1979*). Cell nuclei were detected using the background signal of remaining free GFP-53BP1 protein by Otsu thresholding[63]. This initial detection was used to create a mask, where objects were filtered for their size, signal saturation, and shape. A band-pass filter was used to enhance the signal generated by objects the size of a 53BP1 focus. Local maxima were then detected using a threshold automatically calculated for each nucleus.

## Mitochondria staining and imaging

Sorted multinucleated H226 cells were stained 2 and 4 days after their isolation. Mitotracker green FM (Thermofisher Scientific, M7514) was used at 150 mM for 20 min, followed by a 5-min incubation in Hoechst 33342 to stain nuclei, and WGA-alexa 647 to stain plasma membranes. Images were acquired with a 60 × 1.35 NA objective.

## Imaging

Cell selection and CLaP were performed on an Olympus IX71 microscope (Olympus Corp.) with the appropriate epifluorescence filters, in medium at 37°C, 5% $CO_2$, with a 10 × 0.4 NA objective and an Orca Flash 4.0 camera (Hamamatsu Photonics).

Images of irradiated GFP-53BP1 expressing cells were taken at two time points using a 40X, 0.85NA objective and compared to identify outliers. Since laser tagging was performed with a 10 × 0.4 NA objective, cells were identified in a new live image at different magnification during tagging.

Automatic acquisition of immunostained samples for characterization of large numbers of cells from purified cell populations was performed with an automated Zeiss AxioObserver Z1 Epifluorescence microscope, at room temperature in PBS with Zen Blue software and a 20 × 0.85 NA objective.

## Cell cycle analysis

Exponentially growing cell cultures were trypsinized, fixed with 70% ethanol, and stored at −20°C until use. Fixed cells were washed with PBS and treated with 0.5% triton X-100 for 10 min at room temperature. After washing with PBS, cells were resuspended in PBS containing 2 µg/mL propidium iodide and 0.2 mg/mL RNase A and incubated for 30 min at room temperature. Samples were analysed by flow cytometry on a FACSCalibur instrument (Becton-Dickinson). Data was analyzed with FlowJo v10 software, and cell cycle phases were determined using the Watson algorithm.

## Conditioned medium

U2OS cells were plated at a density of 2 million cells per 10 cm dish. 24 hr later, medium was removed and filtered through a 0.2 µm filter to ensure sterility and remove any floating cells. Conditioned medium was always prepared fresh.

## Polymerase chain reaction

10 cells were resuspended in 40 µL of water and boiled for 10 min. Samples were subjected to 24 PCR cycles using Agilent Herculase II with primer sets specific for the mitochondrial gene *Cytb* of either dog or mouse. 2 µL of each reaction were then used for PCR or qPCR with each primer set. Total genomic DNA from either dog or mouse cells were used as controls. The primers used are *Cytb1L*(5'- CATAGCCACAGCATTCATGG −3'), *Cytb1R*(5'- GGATCCGGTTTCGTGTAGAA −3'), and *Cytb2L*(5'- CCTCAAAGCAACGAAGCCTA −3'), *Cytb2R*(5'- TCTTCGATAATTCCTGAGATTGG −3'), which amplify fragments of 247 nt and 196 nt from the *Cytb* gene of dog and mouse, respectively. Quantitative PCR was performed with the above primer pairs using the 2X SYBR Green Master Mix (Bimake) and an ABI7500 instrument (ThermoFisher). The amount of dog and mouse DNA in each sample was calculated using standard curves made from serial dilutions of genomic DNA isolated from each cell type.

## Immunoblotting

Immunoblotting was performed with total cellular extract using standard protocols. Antibodies used were rabbit anti-53BP1 (Santa-Cruz, sc-22760) and rat anti-tubulin (Abcam, ab6161).

# Acknowledgements

We thank Maryam Tabatabaei and Sébastien Talbot (Université de Montréal) for help with experiments using DRG cells. This work was supported by grants from the Natural Science and Engineering Research Council of Canada to ED, EBA, SC and HW, Genome Canada/Génome Québec and Canadian Cancer Society to SC, Fonds de Recherche du Québec – Nature et Technologies to SC, and the Canadian Institutes of Health Research to ED, EBA and HW SC, EBA and HW hold salary awards from the Fonds de Recherche du Québec – Santé.

# Additional information

## Funding

| Funder | Author |
| --- | --- |
| Canadian Institutes of Health Research | El Bachir Affar<br>Elliot Drobetsky<br>Hugo Wurtele |
| Natural Sciences and Engineering Research Council of Canada | El Bachir Affar<br>Elliot Drobetsky<br>Hugo Wurtele<br>Santiago Costantino |
| Fonds de Recherche du Québec - Santé | El Bachir Affar<br>Hugo Wurtele |
| Genome Canada | Santiago Costantino |
| Canadian Cancer Society | Santiago Costantino |

| Fonds de Recherche du Québec - Nature et Technologies | Santiago Costantino |

The funders had no role in study design, data collection and interpretation, or the decision to submit the work for publication.

## Author contributions

Loïc Binan, Conceptualization, Data curation, Software, Formal analysis, Investigation, Methodology, Writing—original draft, Writing—review and editing; François Bélanger, Investigation, Methodology, Writing—review and editing; Maxime Uriarte, Jean François Lemay, Jean Christophe Pelletier De Koninck, Joannie Roy, Data curation, Investigation, Writing—review and editing; El Bachir Affar, Resources, Methodology, Writing—review and editing; Elliot Drobetsky, Resources, Funding acquisition, Investigation, Writing—original draft, Writing—review and editing; Hugo Wurtele, Resources, Formal analysis, Funding acquisition, Investigation, Writing—review and editing; Santiago Costantino, Conceptualization, Resources, Data curation, Software, Formal analysis, Supervision, Funding acquisition, Investigation, Methodology, Writing—original draft, Project administration, Writing—review and editing

## Author ORCIDs

Jean François Lemay http://orcid.org/0000-0002-3540-1627
Santiago Costantino http://orcid.org/0000-0002-2454-2635

## Decision letter and Author response

Decision letter https://doi.org/10.7554/eLife.45239.017
Author response https://doi.org/10.7554/eLife.45239.018

## Additional files

### Supplementary files

• Transparent reporting form
DOI: https://doi.org/10.7554/eLife.45239.015

### Data availability

All data generated or analysed during this study are included in the manuscript and supporting files.

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
