## [Decision Letter]

Thank you for submitting your article "Opto-magnetic capture of individual cells based on visual phenotypes" for consideration by *eLife*. Your article has been reviewed by three peer reviewers, one of whom is a member of our Board of Reviewing Editors, and the evaluation has been overseen by Anna Akhmanova as the Senior Editor. The following individual involved in review of your submission has agreed to reveal their identity: Paul S Maddox (Reviewer #3).

The reviewers have discussed the reviews with one another and the Reviewing Editor has drafted this decision to help you prepare a revised submission.

Summary:

In this manuscript, the authors present a follow-up to their single-cell laser tagging system (named CLaP) where they couple live-cell labelling with Biotin-4-fluorescein with magnetic cell retrieval using ferromagnetic streptavidin-coated beads. They authors show convincingly that the method has high selectivity and efficiency. They highlight the potential of this approach by isolating cells with unusual phenotypes such as fast-resolver of 53BP1 IR-induced foci as well as multinucleated cells. They show that cells tolerate the isolation procedure and can be expanded afterwards.

All of the reviewers were enthusiastic about the concepts, and felt the potential benefits of the method to a broad range of applications in the community made the study potentially interesting. However, the reviewers raised a number of points that would require addressing before the study can be further considered for publication.

Major points:

- The benefit of isolating artificially induced multi-nucleated cells was not very clearly made and Figure 6 and Figure 6— figure supplement 1 were not convincing or informative as to the benefit of using this approach. It would be important to make this clearer and provide additional rationale and evidence for why this application of scMOCa is of importance to the broader scientific community.

- It would be important to show the approach works for more sensitive cells where the method will likely be most useful. It would be very helpful to show that more sensitive cells that do not usually tolerate FACS well, such as stem cells or primary neurons, can be isolated using this approach and survive in culture post-isolation.

- Please define how critical the collection chamber setup shown in Figure 1C is. Could one simply trypsinise the cells and pass them on a MACS-type column?

- Can scMOCa be used on fixed cells (e.g. after immunofluorescence)? This would open up additional applications.

---

## [Author Response]

Major points:- The benefit of isolating artificially induced multi-nucleated cells was not very clearly made and Figures 6 and Supp Figure 6—figure supplement 1 were not convincing or informative as to the benefit of using this approach. It would be important to make this clearer and provide additional rationale and evidence for why this application of scMOCa is of importance to the broader scientific community.

We agree and have successfully repeated our experiments without any artificial induction. We originally induced polyploidy to increase our chance of detecting various populations of polyploid cells (4N, 8N, 16N.). For this resubmission, we were able to capture spontaneously arising multinucleated cells (which arise in very low numbers) and culture them for several days. Since these cells do not divide, we stained them using Mitotracker green to show they retain metabolic activity. Multinucleated cells have recently garnered considerable interest in light of burgeoning evidence that these play important roles in drug resistance and cancer relapse. Recent studies suggest these cells persist in tumour tissues after treatment. They normally do not divide but may later undergo a budding process resulting in de novo formation of mononucleated cells that contribute to cancer relapse. They also transmit stemness promoting factors and sub-genomes to neighboring cells. To the best of our knowledge live multinucleated cells cannot be rigorously sorted by classical methods such as FACS, since e.g., cells in G2 cannot be distinguished from tetraploid cells in G1. As such, isolation of these rare cells would depend solely on visual identification. We modified the manuscript to better explain the interest of sorting such cells.

- It would be important to show the approach works for more sensitive cells where the method will likely be most useful. It would be very helpful to show that more sensitive cells that do not usually tolerate FACS well, such as stem cells or primary neurons, can be isolated using this approach and survive in culture post-isolation.

To address this, we show (subsection “Rare cells can be sorted and expanded with high efficiency and specificity”) that scMOCA can be successfully used for sorting 3 primary cell types: lung fibroblasts (Figure 4A), endothelial cells (Figure 4B), and neurons (Figure 4C).

In addition we also added experiments (see the aforementioned subsection) with mouse embryonic stem cells. In the new version of Figure 4 we show that these cells, known to be remarkably fragile and do not tolerate FACS well, can be captured with scMOCA and kept proliferating for at least 15 days (Figure 4D).

- Please define how critical the collection chamber setup shown in Figure 1C is. Could one simply trypsinise the cells and pass them on a MACS-type column?

To address this, we repeated the same type of experiment we previously used to characterise the efficacy of scMOCa, i.e. cells were tagged with the laser but this time we used MACS columns for capture. We provide the quantification of this experiment and show that approximately 75% of cells captured by columns are contaminating cells that do not have any magnetic bead attached. Moreover, this procedure yields important losses, as only an average of 17% of the positive cells were captured in three independent assays. These poor results can be explained by the fact that MACS columns are optimised for larger cell samples, where the target population to be extracted represents a more important proportion of the original sample and therefore contamination with negative cells may not represent a major drawback. This result underscores the importance of our design that significantly limits cell manipulations such as dilution and spinning, while also reducing the influence of turbulence and flow forces that contaminate the collection tubes with negative cells. The manuscript was modified in the subsection “Rare cells can be sorted and expanded with high efficiency and specificity” to clarify these points.

- Can scMOCa be used on fixed cells (e.g. after immunofluorescence)? This would open up additional applications.

This is an interesting possibility. However, we strongly believe that the primary strength of our technology lies in the ability to manipulate live cells. Other commercially available technologies, such as laser microdissection, can do this adequately with fixed cells.

Nevertheless, we have successfully performed the suggested experiments with fixed cells and were able to attach magnetic beads on chosen cells as well as detaching with trypsin and capturing them. Importantly, there is a limitation in the purity of the sample that can be achieved as fixed and crosslinked cells are often attached to negative neighboring cells which can be captured. The use of scMOCa with fixed cells would therefore be restricted to very low-density cell cultures.

In addition, we confronted a major challenge in manipulating fixed and permeabilized cells for immunofluorescence. Since cell membrane integrity is severely compromised during these protocols, even if beads attach to target cells, trypsin and magnetic forces often shear cells, potentially resulting in capture of severely physically-damaged cells.

In view of the above, we feel that our technology is not yet optimized for capture of fixed cells.